# Enhanced Astrocyte Activity and Excitatory Synaptic Function in the Hippocampus of Pentylenetetrazole Kindling Model of Epilepsy

**DOI:** 10.3390/ijms241914506

**Published:** 2023-09-25

**Authors:** Franco Díaz, Freddy Aguilar, Mario Wellmann, Andrés Martorell, Camila González-Arancibia, Lorena Chacana-Véliz, Ignacio Negrón-Oyarzo, Andrés E. Chávez, Marco Fuenzalida, Francisco Nualart, Ramón Sotomayor-Zárate, Christian Bonansco

**Affiliations:** 1Centro de Neurobiología y Fisiopatología Integrativa (CENFI), Instituto de Fisiología, Facultad de Ciencias, Universidad de Valparaíso, Valparaíso 2360102, Chile; franco.diazg@gmail.com (F.D.); freddy.aguilar@postgrado.uv.cl (F.A.); mario.wellmann@gmail.com (M.W.); andres.martorellh@gmail.com (A.M.); camila.gonzaran@gmail.com (C.G.-A.); lorena.ch89@gmail.com (L.C.-V.); ignacio.negron@uv.cl (I.N.-O.); marco.fuenzalida@uv.cl (M.F.); 2Escuela de Ciencias de la Salud, Universidad Viña del Mar, Viña del Mar 2580022, Chile; 3Programa de Magíster en Ciencias Biológicas mención Neurociencia, Facultad de Ciencias, Universidad de Valparaíso, Valparaíso 2360102, Chile; 4Escuela de Fonoaudiología, Facultad de Salud, Universidad Santo Tomás, Viña del Mar 2561780, Chile; 5Centro Interdisciplinario de Neurociencia de Valparaíso (CINV), Instituto de Neurociencia, Facultad de Ciencias, Universidad de Valparaíso, Valparaíso 2360102, Chile; andres.chavez@uv.cl; 6Laboratory of Neurobiology and Stem Cells, NeuroCellT, Department of Cellular Biology, Faculty of Biological Sciences, University of Concepcion, Concepción 4070386, Chile; frnualart@udec.cl; 7Center for Advanced Microscopy CMA BIOBIO, Faculty of Biological Sciences, University of Concepcion, Concepción 4070386, Chile

**Keywords:** astrogliosis, gliotransmission, excitation–inhibition imbalance, release probability, PTZ-induced kindling, epilepsy

## Abstract

Epilepsy is a chronic condition characterized by recurrent spontaneous seizures. The interaction between astrocytes and neurons has been suggested to play a role in the abnormal neuronal activity observed in epilepsy. However, the exact way astrocytes influence neuronal activity in the epileptogenic brain remains unclear. Here, using the PTZ-induced kindling mouse model, we evaluated the interaction between astrocyte and synaptic function by measuring astrocytic Ca^2+^ activity, neuronal excitability, and the excitatory/inhibitory balance in the hippocampus. Compared to control mice, hippocampal slices from PTZ-kindled mice displayed an increase in glial fibrillary acidic protein (GFAP) levels and an abnormal pattern of intracellular Ca^2+^-oscillations, characterized by an increased frequency of prolonged spontaneous transients. PTZ-kindled hippocampal slices also showed an increase in the E/I ratio towards excitation, likely resulting from an augmented release probability of excitatory inputs without affecting inhibitory synapses. Notably, the alterations in the release probability seen in PTZ-kindled slices can be recovered by reducing astrocyte hyperactivity with the reversible toxin fluorocitrate. This suggests that astroglial hyper-reactivity enhances excitatory synaptic transmission, thereby impacting the E/I balance in the hippocampus. Altogether, our findings support the notion that abnormal astrocyte–neuron interactions are pivotal mechanisms in epileptogenesis.

## 1. Introduction

Epileptogenesis is a process by which normal brain tissue is transformed into a cellular region prone to generating a crisis state or seizure due to a progressive increase in electrical activity, which can reach a continuous convulsive state or status epilepticus [1,2]. While the main cellular and molecular changes that lead to those alterations have been associated with changes in neuronal activity and synaptic functions, increasing evidence suggests an essential contribution of astrocyte-to-neuron signaling in the mechanisms that underlay seizure activity initiation, propagation, and recurrence [3,4,5,6]. Indeed, in both mesial temporal lobule epilepsy patients and chronic epilepsy animal models, astrocytes have been shown to undergo morphological, molecular, and functional transformation giving rise to astrogliosis, malfunction of the glutamate transporter, over-expressing hemichannels, as well as alteration of glutamatergic and purinergic receptors among other proteins implicated in astrocyte-to-neuron signaling [6,7,8,9,10].

Kindling is a chronic model of epilepsy where repetitive and intermittent administration of sub-convulsant chemical or electrical stimuli can lead to progressive amplification of seizures, culminating in generalized seizure activity [4,6,9,10]. Experimental evidence suggests that reactive astrocytes from electrical-kindled rats display Ca^2+^-mediated hyper-excitability, increasing glutamate-mediated gliotransmission [6,11,12,13]. Consequently, glutamate released from astrocytes activates metabotropic glutamate receptors (mGluR1/5 subtypes) to increase glutamate release from presynaptic terminals [10]. This mechanism has been proposed as an excitatory astrocyte-to-neuron loop that raises the excitatory transmission required to reach the epileptic seizure threshold, contributing to the generation of recurrent episodes during epileptogenesis [6,10,14]. For instance, in electrical-kindling rats, alterations have been reported associated with rising gliotransmission and glutamatergic tone, suggesting that the alteration of astroglial signaling might be an important mechanism underlying the epileptogenesis process [15,16,17]. However, whether chronic treatment with pentylenetetrazole (PTZ), a non-competitive GABA_A_ receptor antagonist widely used to induce epilepsy model [16,17], induced abnormal astrocytic signaling that affects neuronal excitability and release probability at CA3-CA1 glutamatergic synapses in the hippocampus remains unknown. Here, we sought to address these issues by recording astrocytic Ca^2+^ patterns and synaptic function from CA1 pyramidal neurons in acute hippocampal slices from the PTZ-kindling mouse model.

## 2. Results

### 2.1. In Vivo Epileptogenic Progression the in PTZ-Kindling Model of Epilepsy

To evaluate whether PTZ-kindled mice display abnormal astrocyte reactivity that could affect neuronal synaptic function in the hippocampus, first, we corroborate that chronic treatment with PTZ induces epileptic state through behavioral evolution towards tonic–clonic seizures according to the Racine scale [9,18,19]. As previously demonstrated [15,19,20], after repeated administration of PTZ, a progression of the epileptic condition was observed (Figure 1A, left panel) and protracted duration of Racine’s stage 5 (Figure 1A, right panel). In this context, two successive stage 5 seizure after PTZ injection (Figure 1A, above) was considered a complete kindled state [21]. These mice were euthanized to obtain the slices for ex vivo and in vitro experiments. In addition, we corroborate the epileptogenic effect generated by chronic administration of PTZ through in vivo brain electrical activity (see Methods; Figure 1B). An increase in local field potential (LFP) displayed as after-discharges (ADs) was observed after the last PTZ injection (Figure 1C). Intrinsic features of the ictal crisis showed a higher power at the low-frequency band (1–3 Hz) at day 13 (D13) compared to D1 (Figure 1D, see Appendix A). Moreover, the number of ADs (e.g., incidence min^−1^) and ADs duration were also higher at D13 than at D1, whereas the latency time for seizures was lower at D13 than at D1 (Figure 1D, see Appendix A). Altogether these in vivo results confirm that, in our hands, chronic administration of PTZ for 13 days generates a PTZ-kindling model of epilepsy.

### 2.2. Astrogliosis and Astrocyte Calcium Wave in the Hippocampus of the PTZ-Kindling Model

Compared to control or vehicle-injected mice, we found that the hippocampus of the PTZ-kindling model displays an abnormal astrocyte reactivity as evidenced by the increase in glial fibrillary acidic protein (GFAP) levels (Figure 2A, see Appendix A), a hallmark of astroglial activation [22]. Likewise, a significant increase in the protein expression of GFAP was also observed in the ventral hippocampus (Figure 2B, see Appendix A), strongly supporting the idea that astroglial dysfunction is considered a cellular phenotype in the epileptic brain [23,24].

Next, we determined whether astroglial Ca^2+^ waves are altered in the hippocampus of the PTZ-kindling model, as previously demonstrated in electrical kindling [6,10,25], by measuring spontaneous intracellular Ca^2+^ elevations from control and kindled hippocampal slices pre-incubated with the astrocyte-specific marker SR101 and the Ca^2+^ probe Fluo-4-AM [6] (see Methods; Figure 3A). A significant difference in the duration of Ca^2+^ transients above the 75th percentile of the cumulative distribution with 23.1 s as a cut-off criterion to sort Ca^2+^ waves into slow (ST) and fast (FT) transients was observed in PTZ-kindling slices (Figure 3B). An increase in duration and oscillation of the Ca^2+^ signal was also observed in PTZ-kindling slices compared to control slices (Figure 3D, see Appendix A). Importantly, the FT/ST index values were lower in epileptic than control or vehicle-injected mice (Figure 3D, see Appendix A), indicating an increase in astroglial Ca^2+^-dependent excitability with an elevated occurrence of slow Ca^2+^ transients. Such increases in astrocyte Ca^2+^-dependent excitability indicate that increased gliotransmission of glutamate from hyperexcitable astrocytes may enhance excitatory neurotransmission within the hippocampus of PTZ-kindling slices. This observation is consistent with prior findings in other kindling epileptic models, suggesting a potential mechanism for the up-regulation of excitatory neurotransmission in the epilepsy [6].

### 2.3. Excitatory Synaptic and Neuronal Excitability Function Is Enhanced in PTZ-Kindling Model

To determine whether astrocyte hyperexcitability alters neurotransmitter release in the hippocampus of the PTZ-kindling model, we monitored basal excitatory synaptic transmission at Schaffer collateral to CA1 synapses in the hippocampal region and detected a significant increase in the frequency but not in the amplitude of spontaneous excitatory postsynaptic currents (sEPSCs; Figure 4A, see Appendix A). In contrast, at synapses formed by interneurons onto CA1 pyramidal neurons, we detected no change in the spontaneous inhibitory postsynaptic currents (sIPSCs; Figure 4B, see Appendix A). The strong increase in the frequency of sEPSC in the PTZ-kindling model could result from an increase in presynaptic excitability and/or release probability at excitatory synapses [10,25]. To test changes in excitability, we measured isolated field excitatory postsynaptic potentials (fEPSPs; see Methods) input/output curves and found a 2-fold enhancement in the EPSP amplitudes at all stimulus intensities in PTZ-kindling model compared to control or vehicle-injected mice (Figure 5A, see Appendix A). Moreover, we found that stimulation intensity required to induce population spike threshold, reliably used to indicate postsynaptic excitability [26], was lower in the PTZ-kindling model than in control and vehicle-injected mice (Figure 5B, see Appendix A), suggesting that neuronal excitability is altered in the hippocampus of PTZ-kindling model. Next, we tested changes in release probability by examining paired-pulse facilitation at excitatory CA3-CA1 synapse and found a significant reduction in the paired-pulse ratio in the PTZ-kindling model compared to control or vehicle-injected mice (Figure 5C, see Appendix A), whereas at inhibitory synapses, paired-pulse depression remains unchanged (Figure 5C, see Appendix A). These findings indicate that in the PTZ-kindling model, astrocyte hyperexcitability alters neuronal excitability and excitatory release probability without affecting inhibitory synaptic function. Moreover, these findings suggest that a change in the excitatory/inhibitory (E/I) synaptic balance might occur in the PTZ-kindling model. To test this possibility, we recorded CA1 pyramidal cells at −40 mV and stimulated monosynaptic excitatory afferents and disynaptic inhibitory pathways to obtain both EPSCs and IPSCs at the same time. Under these experimental conditions, we found that the excitatory-to-inhibitory ratio was higher in the PTZ-kindling model compared to control or vehicle-injected mice (Figure 5C, see Appendix A), indicating that abnormal astrocyte function impacts synaptic function towards enhancement of excitatory transmission and neuronal excitability in the epileptic brain of PTZ-kindling model.

### 2.4. Metabolic Arrest of Abnormal Astrocyte Restores the Release Probability at Excitatory Synapse in the PTZ-Kindling Model

Previous evidence indicates that the reversible toxin fluorocitrate (FC) decreases the astrocytic metabolism [27], producing a reduction in the Ca^2+^-mediated glutamate release from astrocytes, increasing paired-pulse ratio, and decreasing release probability at CA3-CA1 synapses in normal conditions [4,28,29]. To test whether FC can revert the increase in release probability observed in PTZ-kindled mice, we test the effects of acute perfusion of FC (200 μM) on fEPSP paired-pulse ratio (PPR). As previously reported [4], FC induces an increase in the PPR in slices from vehicle-injected mice (Figure 6). Notably, in PTZ-kindled slices, where the PPR is known to be reduced (Figure 5B and Figure 6, see Appendix A), acute application of FC significantly increased PPR (Figure 6, see Appendix A) to similar values of PPR from vehicle-injected slice (Figure 6). These results show that by inhibiting astrocytic metabolism FC effectively counteracts neuronal hyper-excitability induced by PTZ, recovering the normal release probability at excitatory synapses and strongly supporting the notion that abnormal astrocyte–neuron communication plays an essential role in epileptogenesis.

## 3. Discussion

Here, we show that PTZ-kindled mice exhibit reactive astrogliosis and an abnormal pattern of Ca^2+^ waves in the hippocampus. The astroglial hyper-excitability likely up-regulates excitatory transmission by increasing the probability of glutamate release at Schaffer collaterals-CA1 synapses. We also found that these changes are synapse-specific, affecting only excitatory but not inhibitory synapses onto CA1 pyramidal neurons, thus, altering the excitatory/inhibitory balance required for proper hippocampal function. The increase in release probability induced by astroglial hyper-excitability can be rescued by arresting astrocyte metabolism, strongly suggesting that a dysregulation of the astrocyte–neuron signaling is implicated in the epileptogenesis induced by chronic treatment with PTZ. Importantly, in other chronic epilepsy models—including electrical kindling in rats—a similar impairment of astrocyte–neuron signaling has been demonstrated [6,9,10], supporting the idea that astrocyte–neuron interaction is an important factor in the pathogenesis of epilepsy.

### 3.1. Ca^2+^-Dependent Astroglial Hyper-Excitability in the Epileptic Hippocampus

Although PTZ-induced chemical kindling is a well-characterized model of the chronic epilepsy [19,21] with considerable evidence for changes at the molecular, cellular, and behavioral stages, our data provide for the first time, evidence that astroglial modulation of synaptic efficacy could be an important step in the epileptic progression induced by PTZ. As previously reported in different animal models [30,31,32], the hippocampal formation of the PTZ-kindling model showed strong and widespread reactive astrogliosis and a higher incidence of slow spontaneous elevations of Ca^2+^ transient. Such abnormal pattern of Ca^2+^ transient has been previously described in electrical kindling epileptic model in rats [10] and in resected tissue from TLE patients [24], suggesting a common mechanism underlying epileptogenesis. Moreover, the increase in the slow astroglial Ca^2+^ transients appears to be associated with an increase in glutamate released from astrocytes (gliotransmission), upregulating glutamate release from presynaptic terminals. Whether the increase in slow astroglia Ca^2+^ transients is responsible for the higher basal glutamate levels reported in patients with TLE epileptogenic foci [33] remains unknown.

Interestingly, purinergic P2Y1 receptors located on astrocytes are overexpressed in the epilepsy [12,34,35], and recently, we have shown that its blocked diminished slow Ca^2+^ transient in astrocyte [6] opens the possibility that the increase in astroglia Ca^2+^ transients in PTZ-kindling model could be due in part to an increase in P2Y1 receptors function. For instance, in electric-kindled rats [6], ATP release through Pannexin-1 from reactive astrocytes is required for P2Y1-induced up-regulation of excitatory synaptic function. Further investigations will be necessary to determine whether this purinergic pathway is implicated in the astroglial hyperactivity of the PTZ-kindled model.

### 3.2. Increase in Excitatory Synaptic Activity Yields E/I Unbalances in the Hippocampal Network of PTZ-Kindled Mice

Normal brain function requires the subtle adjustment between synaptic excitation and inhibition [36]. Here, we described that chronically epileptic PTZ-kindled mice showed a two-fold increase in the E/I ratio of hippocampal CA1 pyramidal cells towards excitation. This was corroborated by increases in the frequencies of excitatory spontaneous activity. A similar E/I increase has been described in the pilocarpine-induced status epilepticus [37]; however, the increase in the E/I ratio is associated with a reduction in the release probability of GABAergic inputs (i.e., minor miniature IPSC frequencies). While these differences could be attributed to the different models used, we propose that in chronic epilepsy, PTZ-induced dysregulation of E/I synaptic balance contributed to the upregulation of input/output function and hyper-excitability of CA1 pyramidal neurons. The origin of this synaptic imbalance that predisposes the healthy brain to transform into an epileptic one has been associated with altering the mechanisms that modulate synaptic transmission and neuronal excitability. One of the well-known modulatory mechanisms is the astroglial signaling mediated by the gliotransmitters [38,39]. Certainly, evidence showed that disrupted gliotransmitter-mediated astrocyte–neuron signaling by reactive astrocytes contributes to synaptic imbalance and hyper-excitability in epileptic circuits [8,24]. In several structures, including the hippocampus, the synaptic activity or autocrine astrocytic signals can activate astroglial Ca^2+^ waves that, in turn, release gliotransmitters, activating receptors that depolarize the membrane potential, reducing the membrane resistance or changing conductance of voltage-dependent channels [24,40,41]. Likewise, gliotransmitters modify glutamatergic and GABAergic synaptic transmission at pre- and postsynaptic levels [4,38,41]. Here, we demonstrated that as in the electrical kindling model [10] chronic treatment with PTZ also induces hyper-excitable reactive astrocytes affecting the modulation of glutamate neurotransmission. The signaling pathways and circuital mechanisms produced by this synaptic imbalance remain an open question.

## 4. Materials and Methods

### 4.1. Animals

Male C57BL/6 strain mice between 30 and 60 days of postnatal age (P30-P60) were obtained from the Animal House Facility at the Faculty of Science, Universidad de Valparaíso. A total of 58 mice were divided into three experimental groups: (i) the control or wild-type group (*n* = 19), (ii) the vehicle group injected with a saline solution (1 mL/kg, intraperitoneally (i.p.), *n* = 14), and (iii) the PTZ group (35 mg/kg, i.p., *n* = 25). All animals were maintained in temperature, humidity, and light-controlled rooms, with food and water ad libitum. Animal handling and use were in strict accordance with the National Institute of Health (USA) guidelines for the use of experimental animals, and the Institutional Animal Ethics Committee approved all the protocols at the Universidad de Valparaíso (Act. N° CBC 34/2022).

### 4.2. Tracing Epileptogenesis in the PTZ-Kindling Mouse Model

PTZ-kindling-induced epilepsy model was generated as previously described [16,21]. Briefly, PTZ was freshly dissolved in saline (0.01 g/mL) solution and sub-convulsive doses (35 mg/kg, i.p.) were injected every other day for 13 days (7 injections, Figure 1A). This protocol led to most animals reaching consecutive Racine stage 5 seizures, while a smaller percentage of animals (<30%) required two additional injections under the same administration schedule. To assess the development of epileptiform activity induced by the PTZ-kindling protocol in vivo, local field potential (LFP) was recorded in freely moving mice undergoing stereotaxic surgery under anesthesia (4% isoflurane; 100% O_2_). Three tungsten microwire electrodes (50 mm diameter, polyamide insulated, California Fine Wires Co, Grover Beach, CA, USA) were implanted intracranially in the right hemisphere at stereotaxic coordinates targeting the motor cortex (2.10 mm AP, 2.0 mm ML from Bregma). Electrodes were then connected to an amplifier (model RHD2132 connected to RHD2000 evaluation system; Intan Tech, Los Angeles, CA, USA). Following surgery, animals were housed individually in a temperature and humidity-controlled room (22 ± 1 °C, 60%, respectively) with access to food and water ad libitum. After the final PTZ injection, behavioral and electrical brain activities were monitored. Electrical brain signals were amplified, digitalized, and filtered (×1000 times; sampling rate: 20 kHz; 0.5–5000 Hz, respectively). Offline LFP analysis was performed using MATLAB software, Version 9.9 (The Mathworks Inc., Natick, MA, USA). LFP was sampled at 1000 Hz and band-pass filtered at 0.1–100 Hz. Interictal spikes and epileptic seizures were manually identified. Interictal spikes were defined as transient high amplitude deflections or poly-spike complexes lasting 20–70 ms, whereas ictal activity (i.e., epileptic seizures) was identified as after discharge (AD), characterized by a spike-wave pattern that appeared with variable latency after each injection. At the most severe stages, seizures were manifested as tonic–clonic episodes, which coincided temporarily with single or multiple ADs of longer duration and amplitude, following ictal depression that ended the seizure. Power spectral density (PSD) was computed using multitaper analysis from the Chronux toolbox (http://www.chronux.org, accessed on 1 March 2023). For PSD analysis, field potentials were divided into 4000 ms segments with 500 ms overlap and a time-bandwidth product of 5 and 9 tapers.

### 4.3. Electrophysiology in Hippocampal Slices

Acute transverse hippocampal slices were obtained from mice of each experimental group as previously described. Briefly, mice were anesthetized with isoflurane, decapitated, and the brain was rapidly removed and placed in ice-cold (<4 °C) artificial cerebrospinal fluid (ACSF) containing (in mM) 124.0 NaCl, 2.7 KCl, 1.25 KH_2_PO_4,_ 2.0 Mg_2_SO_4_, 26.0 NaHCO_3_, 2.5 CaCl_2_, and 10.0 glucose equilibrated with 95% O_2_ and 5% CO_2_ (pH 7.4). Hippocampal slices (300–350 μm thick) were obtained using a Vibroslice microtome (VSL, WPI, Sarasota, FL, USA) and were incubated for at least 1 h in ACSF at room temperature (21–24 °C). Then, slices were transferred to an immersion-recording chamber (2 mL) and fixed to an upright microscope stage (FN100 IR; Nikon Inc., Tokyo, Japan), equipped with infrared and differential interference contrast imaging devices and a 40× water immersion objective. Two electrophysiological recording methods were employed: extracellular field potentials and whole-cell patch-clamp recordings. For field excitatory postsynaptic potentials (fEPSPs), a glass pipette (2–4 MΩ) filled with ACSF was placed in the middle of the CA1 stratum radiatum. This pipette was connected to an AC amplifier (P-5 series; Grass) with a gain of 310,000×, a 3.0-kHz low-pass filter, and a 0.30 Hz high-pass filter. To evoke fEPSPs, a bipolar concentric electrode (concentric platinum/iridium electrode, 125 μm outer diameter, FHC Inc., Bowdoin, ME, USA) was placed in the stratum radiatum within 100–200 μm from the recording site to activate the Schaeffer collateral fibers using bipolar cathodic stimulation (50 μs, 0.3 Hz, 20–100 μA), generated by a stimulator (A.M.P.I., Jerusalem, Israel) connected to an isolation unit (Isoflex, Jerusalem, Israel, AMPI). fEPSP recordings were made in the continuous presence of picrotoxin (PTX, 10 μM), a GABA_A_ antagonist. In a subset of experiments, the astrocyte inhibitor fluorocitrate (FC; 200 μM; Figure 6) [4,42], a reversible toxin used to decrease astrocytic metabolism [27] as inhibited aconitase that prevents the conversion of citrate to isocitrate, reducing glucose metabolism, energy stores, and cellular respiration, was used. Moreover, this concentration reportedly decreases Ca^2+^-dependent glutamate release from astrocytes in the hippocampus [4].

Whole-cell voltage-clamp recordings were made from CA1 pyramidal cells using an EPC-7 amplifier (Heka Instruments, Lambrecht, Germany) through patch-type pipette electrodes (~5 MΩ) containing (in mM) 130 KMeSO_4_, 10 HEPES, 4 ATP-Na_2_, and 20 KCl (290 mOsm, pH 7.3). Excitatory and inhibitory evoked postsynaptic currents (eEPSCs and eIPSCs, respectively) were elicited every 3 s by changing CA1 holding potential at the reversal potential for glutamatergic (i.e., 0 mV) or GABAergic (i.e., −60 mV) synaptic input. Signals were filtered at 3.0 kHz and acquired at 4.0 kHz using an A/D converter (ITC-16; Instrutech, Reutlingen, Germany). The excitatory and inhibitory ratio (E/I ratio) was calculated using the peak amplitude and the total area of evoked compound postsynaptic currents recorded at −40 mV in the continuous presence of the NMDA receptor antagonist APV (25 μM), which includes both excitatory and inhibitory components. All recordings were stored with Pulse FIT software, Version 8.5 (Heka Instruments, Lambrecht, Germany), and cells that exhibited a significant change in access resistance (>20%) were excluded from the analysis. Paired-pulse ratio (PPR) was defined as the ratio of the amplitude of the second response to the amplitude of the first response. PPR was calculated from the control, vehicle, and PTZ-animal models as an indirect measurement of changes in release probability. Offline recording analysis was performed using the pClamp software (Version 10.3) by Molecular Devices (San Jose, CA, USA).

### 4.4. Ca^2+^ Imaging in Astrocytes

Intracellular Ca^2+^ elevations from astrocytes were monitored by fluorescence microscopy using Fluo-4 AM, a cell-permeant Ca^2+^ indicator, as previously described [6]. To confirm the specific recording of Ca^2+^ signals from astrocytes, slices were first incubated with the astroglial morphological marker sulforhodamine-101 (SR101; 0.5–1 μM) [43,44] for approximately 30 min in low Ca^2+^ (0.5 mM)/high Mg^2+^ (4 mM) ACSF at 32–34 °C. Subsequently, the slices were transferred to a chamber with regular ACSF for 30 min and later incubated with Fluo-4 AM (1–2 μL of the dye dissolved in pluronic acid at 0.01% was dropped over the hippocampus, resulting in a final concentration of 5–10 μM) for approximately 75 min in regular ACSF at room temperature. Astrocytes were imaged using a CCD camera (Andor DR328G; Andor Technologies PLC, Belfast, Ireland) attached to the microscope (Nikon, Japan) and controlled by the Niss-Elements AR 3.2 software (Nikon, Tokyo, Japan) that was also used for offline analyses. The fields of measure were randomly selected in the anteroposterior axis of the CA1 stratum radiatum, with astrocytes exhibiting minimum fluorescence values of 2000 units in a Look-Up Table (LUT) implemented by the NISS-Elements software (Version 4.3). Cells were illuminated with a Xenon lamp at 490 nm (200–400 ms exposure; 36,700 mm^2^ area). Images were acquired at 1 Hz for 5 min, regulated by a shutter (Lambda SC-Smart shutter, Sutter Instrument Company, Novato, CA, USA). Analyses of astroglial Ca^2+^ levels were restricted to the cell body, and Ca^2+^ transients were estimated as changes in the fluorescence signal over the baseline (∆F/F0) after background subtraction. For every individual astrocyte, the baseline was obtained by extracting and averaging the fluorescence values of at least 30 consecutive frames, where the astrocyte exhibited no spontaneous Ca^2+^-dependent activity. Changes in fluorescence were considered events when the ∆F/F0 intensity exceeded the fluorescence of the baseline by at least two standard deviations for no less than five consecutive frames. For multi-peak astroglial Ca^2+^ activity, events, where the ∆F/F0 dropped to half of the maximum fluorescence intensity from baseline, were considered independent transients. Additionally, those where the ΔF/F0 decreased below 50% from F0 were treated as distinct events. As previously described [6,25], the duration of astroglial Ca^2+^ transients was distributed among two populations starting from the 75th percentile of the cumulative distribution. That value was employed as a cut-off criterion from which random events in the same astrocyte were classified as fast or low transients (FTs < 23.1 s > STs, respectively). In addition, the FT/ST frequency ratio was included as an index of astroglial excitability, where a lower index would be associated with a greater Ca^2+^-dependent excitability [6].

### 4.5. Western Blot

For protein determinations, ventral hippocampal tissues from 6 animals (both control and PTZ-kindling) were isolated and homogenized using RIPA buffer (pH = 8.0, 150 mM, NaCl, 50 mM Tris-HCl, 1% *v*/*v* Nonidet P40, 0.1% *w*/*v* SDS, 2 mM EDTA, 1.5 mM PMSF) whit protease inhibitor cocktail (Cat# G6521, Promega™). Total protein concentration was determined using the Bio-Rad Protein Assay (Bio-Rad Laboratories, Inc., Richmond, CA, USA) with bovine serum albumin as standard, and the readout was performed at 595 nm in a microplate spectrophotometer (Epoch™, BioTek Instruments Inc., Winooski, VT, USA). Thirty micrograms of total protein from each sample were separated by 10% SDS-PAGE and transferred to nitrocellulose membranes (Cat# 88018, 0.45 μm pore, Thermo Scientific™, Rockford, IL, USA) at 350 mA for 1.5 h. Unspecific membrane binding sites were blocked using 5% skim milk in TTBS (0.1% Tween-20, 20 mM TBS, 137 mM NaCl) for 1 h at room temperature. Later, nitrocellulose membranes were incubated overnight at 4 °C with rabbit anti-GFAP diluted 1:20,000 (Cat# Z033401-2, Aligent Technologies, Santa Clara, CA, USA), and rabbit anti-GAPDH diluted 1:10,000 (Cat# ab9485, Abcam, Cambridge, MA, USA) as the constitutive protein. Membranes were washed with T-TBS and then incubated for 1 h at room temperature in Peroxidase-conjugated AffiniPure F(ab’)2 Fragment Donkey anti-rabbit (Cat# 711-036-152, Jackson Inmuno Research, Laboratories, Inc., West Grove, PA, USA) secondary antibody diluted in blocking solution 1:20,000 for GFAP and GAPDH. For chemiluminescent detection, SuperSignalTM West Dura Extended Duration Substrates (Cat# 34075, Thermo Fisher Scientific, Waltham, MA, USA) were used, and the resulting images of the membranes were obtained using a benchtop transilluminator (Omega Lum G, Aplegen, San Francisco, CA, USA). The images were analyzed using Image-J™ software (http://rsbweb.nih.gov/ij/, accessed on 2 February 2023).

### 4.6. Immunohistochemistry

Male mice were anesthetized with isoflurane using a mask (isoflurane 2% in 0.6 L/min air flow) connected to an animal anesthesia system (model 510, RWD Life Science Co. Ltd., Shenzhen, China). Mice were transcardially perfused with saline (0.9% *w*/*v* NaCl), followed by ice-cold fixative solution (4% *w*/*v* paraformaldehyde in phosphate-buffered saline solution (PBS) 0.1 M with pH 7.4). Brains were removed from the skull and post-fixed for 30 min before the dehydration in 20% *w*/*v* sucrose solution for 48 h at 4 °C. Coronal hippocampal slices were prepared on a cryostat (model KD-2950, Kedee, China) for immunofluorescence. Slices were washed (6 × 5 min) in 0.05 M PBS + 0.5% Triton X-100 (PBS-TX; pH 7.4), followed by incubation overnight with anti-GFAP primary antibodies (monoclonal mouse; 1/200; Merck Chemicon) at room temperature (22–24 °C). Samples were washed (6 × 5 min) with 0.05 M PBS (pH 7.4), incubated with the secondary antibody CY2 (anti-rabbit 1:200; Immunoreasearch Inc., West Grove, PA, USA) for 5 h, washed (6 × 5 min) with 0.05 M PBS (pH 7.4), and mounted using DAKO fluorescent mounting medium (DAKO North America Inc., Santa Clara, CA, USA). Samples were visualized using a Zeiss 780 confocal microscope. Fluorescence (expressed in arbitrary units, AU) and fluorescence area average (% concerning CA1 area) were analyzed using Image J software (https://imagej.net/ij/index.html, accessed on 2 February 2023) (NIH, Bethesda, MD, USA). The total number (*n*) of cells counted for each group was approximately 60 and three fields were selected from every slice. Results are expressed as the relative area of immunostaining (i.e., the size of individual staining units).

### 4.7. Statistical Analysis

After conducting a distribution analysis to determine if the data conformed to a normal distribution (Shapiro–Wilk test, Kolmogorov–Smirnov test), either parametric (Student’s two-tailed *t*-test) or a non-parametric test (Mann–Whitney test) was used. ANOVA or Kruskal–Wallis tests were used as appropriate for multiple comparisons, with post hoc Bonferroni correction. Comparisons regarding incidence, latency, and duration of the ictal crisis were analyzed using Student’s *t*-test. All statistical analyses were conducted using the GraphPad Prism software (Version 8.0.1), with significance set at *p* < 0.05. Unless otherwise stated, all data presented in the figures are depicted as mean ± SEM.

## Figures and Tables

**Figure 1 ijms-24-14506-f001:**
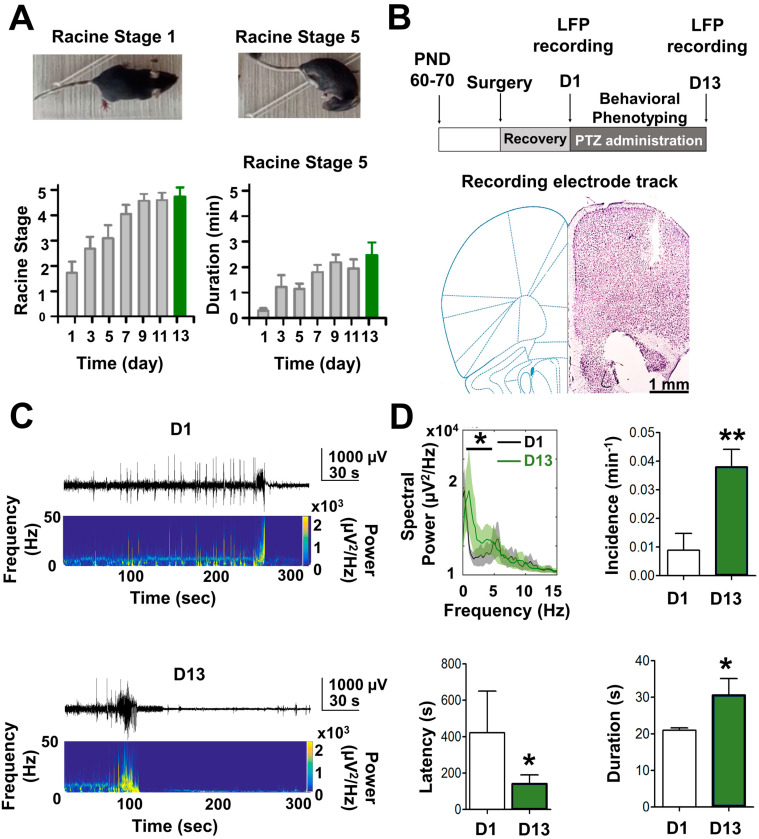
In vivo characterization of the pharmacological kindling model of epilepsy induced by pentylenetetrazole (PTZ). (**A**) Representative images of a mouse in stage 1 (e.g., repetitive movement of the head and neck) and stage 5 (e.g., tonic–clonic seizure) of the Racine scale [19]. The temporal progression of the duration of Racine stage 5 during PTZ (35 mg/kg i.p.) administration is shown (*n* = 5). (**B**) Diagram of the temporal course for experimental protocols (top). After the post-implantation recovery of electrodes, PTZ was administered for 13 days every 48 h (days 1, 3, 5, 7, 9, 11, and 13). Bottom: Coronal section of a mouse brain stained with cresyl violet showing the recording electrode track. (**C**) Representative recording traces of local field potential (LFP) and their respective power spectral density of mice exposed to 1 (D1) and 7 (D13) doses of PTZ. (**D**) Summary graph showing the spectral power frequency, incidence, latency, and duration of after discharges (ADs) recorded at D1 and D13 of pharmacologically kindled mice (*n* = 5) (* *p* < 0.05; ** *p* < 0.01).

**Figure 2 ijms-24-14506-f002:**
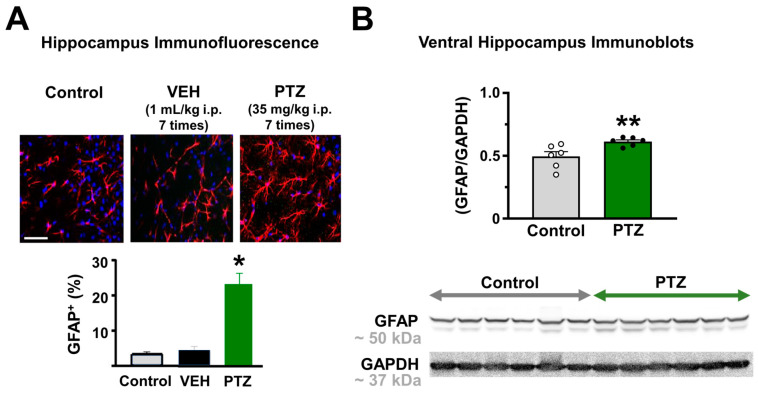
Hippocampal astrogliosis induced by pentylenetetrazole (PTZ) in mice. (**A**) Representative immunofluorescence (top) and summary plot of GFAP staining (bottom) of hippocampal slices from control (*n* = 7), vehicle (VEH, 1 mL/kg i.p. of saline solution; *n* = 8), and PTZ (35 mg/kg i.p. (*n* = 8)) mice showing the immunohistochemical staining of glial fibrillary acidic protein (GFAP, an astrocyte marker) in red and Hoechst staining (DNA marker) in the blue (scale bar = 200 µm). (**B**) Summary graph of GFAP protein levels in the ventral hippocampus for both control (*n* = 6) and PTZ (*n* = 6) mice and original blots for GFAP and GAPDH for both experimental groups (* *p* < 0.05; ** *p* < 0.01).

**Figure 3 ijms-24-14506-f003:**
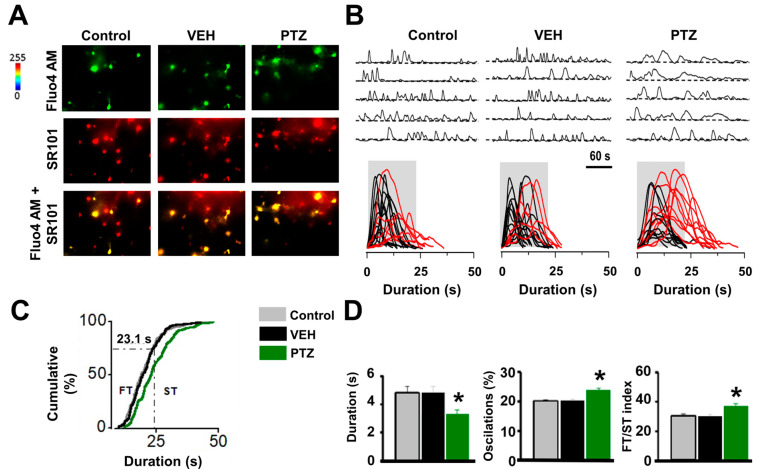
Increases in astrocyte Ca^2+^-dependent excitability in PTZ-kindling mice model. (**A**) Representative fluorescence microscopy images obtained from 5 min video of fluorescence transients for Fluo-4 AM (green), SR101 (red), and co-application of Fluo-4 AM and + SR101 on astrocytes activity in the CA1 stratum radiatum in control, vehicle (VEH, 1 mL/kg i.p. of saline solution), and PTZ-induced (35 mg/kg i.p.) kindled hippocampal slices (calibration bar 20 µm). (**B**) Top: Representative fluorescence traces of spontaneous Ca^2+^ elevations in five astrocytes in the same hippocampal slice from control, vehicle, and PTZ-injected mice. Bottom: Spontaneous Ca^2+^ elevations obtained from several astrocytes (>10 cells) aligned at the start of every transient. Gray boxes show the temporal window within 23.1 s. Representative slow Ca^2+^ transients (STs) exceed the cut-off duration in the gray box (red traces), while the fast transients do not (FTs, black traces). (**C**) Cumulative distribution plot of spontaneous Ca^2+^ events duration obtained from all astrocytes analyzed for control (*n* = 179), vehicle (*n* = 185), and PTZ (*n* = 155). The dashed line indicates the 23.1 s-duration cut-off criterion to classify FT and ST populations. (**D**) Summary graphs of astroglial Ca^2+^ transients mean duration (left), ST percent (middle), and FT/ST ratio (right) for control (*n* = 179, 163 and 163), vehicle (*n* = 185, 180 and 180), and PTZ groups (*n* = 155, 154 and 154) (* *p* < 0.05).

**Figure 4 ijms-24-14506-f004:**
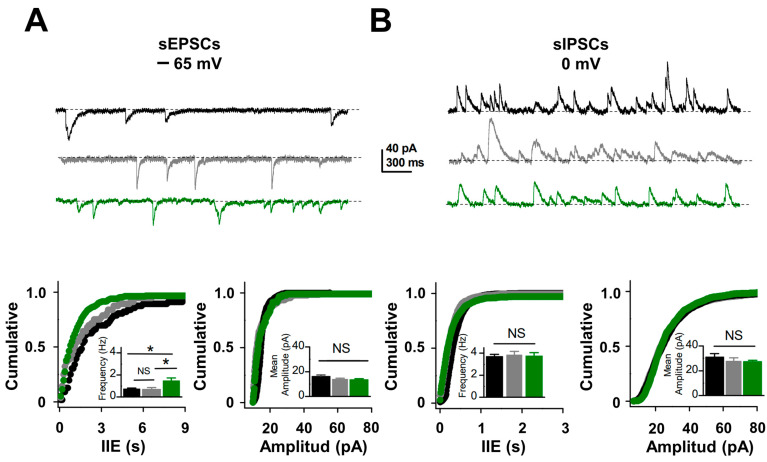
An increase in the frequency of excitatory synaptic events is induced by PTZ in mice. (**A**) Representative traces (top) and summary plots (bottom) of the frequency and amplitude of spontaneous excitatory postsynaptic currents (sEPSCs) recorded from CA1 pyramidal neuron from control (black, *n* = 10), vehicle (gray, VEH, 1 mL/kg i.p. of saline solution, *n* = 10), and PTZ (green, 35 mg/kg i.p., *n* = 14). Note that the frequency of sEPSCs in hippocampal slices from PTZ was significantly higher compared to control or vehicle-injected mice. (**B**) Representative traces (top) and summary plot (bottom) show that basal spontaneous inhibitory postsynaptic currents (sIPSCs) are similar between control (*n* = 18), vehicle (*n* = 12), and PTZ (*n* = 16) groups (* *p* < 0.05; NS, no significant).

**Figure 5 ijms-24-14506-f005:**
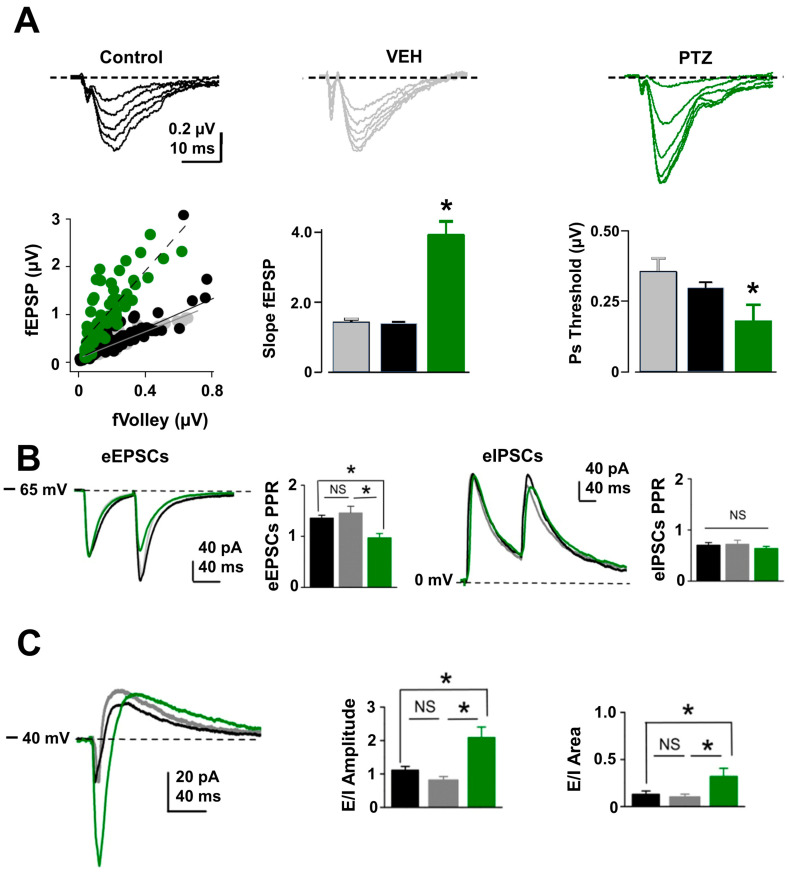
Excitatory/inhibitory ratio is altered in pentylenetetrazole (PTZ)-kindling hippocampal slices. (**A**) (Top) Representative family response of isolated fEPSP traces (in the continuous presence of 10 µM PTX to block inhibitory inputs) from control (black), vehicle (gray, VEH, 1 mL/kg i.p. of saline solution), and PTZ (green, 35 mg/kg i.p.) injected mice. (Bottom) Summary plot showing the fEPSP amplitude vs. the fiber volley (left), the slope of fEPSP (middle), and population spike (Ps; right) threshold in hippocampal slices from control (*n* = 9, 9, 9), vehicle (*n* = 5, 5, 6), and PTZ (*n* = 5, 5, 5). Note that PTZ slices show higher fEPSP amplitude and Ps compared to the control or vehicle group. (**B**) Representative traces of paired-pulse ratio (PPR; 100 ms inter-stimulus intervals) and summary plot showing that PPR of excitatory postsynaptic currents (EPSCs) is decreased in PTZ-kindling (*n* = 14) hippocampal slices compared to control (*n* = 10) or vehicle-injected mice (*n* = 13), whereas PPR of inhibitory postsynaptic currents (IPSCs) is similar between control (*n* = 10), vehicle (*n* = 13), and PTZ (*n* = 14) groups. (**C**) Representative traces of composed postsynaptic currents (cPSCs) were recorded at −40 mV from the same CA1 pyramidal neuron from control, vehicle, and PTZ hippocampal slices. Note that both the amplitude and the area of the excitatory component are higher in PTZ (*n* = 9) vs. control (*n* = 9) or vehicle (*n* = 10) groups, suggesting a change in the E/I ratio (* *p* < 0.05; NS, no significant).

**Figure 6 ijms-24-14506-f006:**
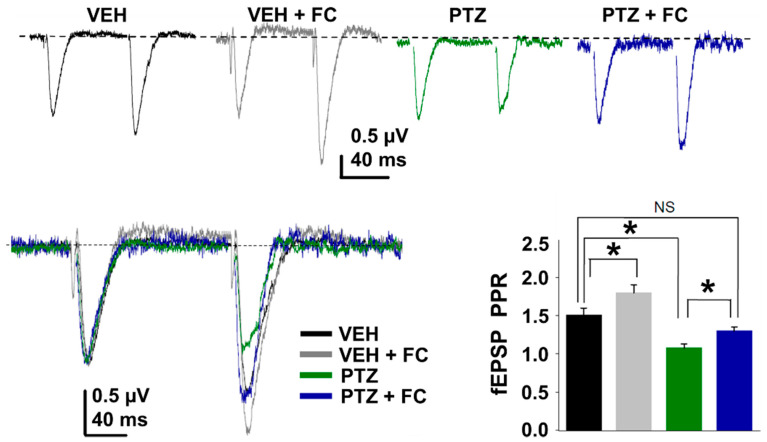
The selective metabolic arrest of astrocytes rescues glutamatergic neurotransmission in PTZ-kindled mice. fEPSP paired-pulse (100 ms inter-stimulus interval) representative traces recorded in the stratum radiatum in the CA1 area of the hippocampus from vehicle (VEH, 1 mL/kg i.p. of saline solution) and PTZ-kindling slices (PTZ, 35 mg/kg i.p.) in the absence and presence of 200 µM fluorocitrate (FC; top). Superimposed traces and summary graph (bottom) showing the effect of FC on PPR in the vehicle (*n* = 12) and PTZ (*n* = 9) groups (* *p* < 0.05; NS, no significant).

## Data Availability

The original contributions presented in the study are included in the article/Appendix A; further inquiries can be directed to the corresponding authors.

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
