# Peer review of "Enhanced Astrocyte Activity and Excitatory Synaptic Function in the Hippocampus of Pentylenetetrazole Kindling Model of Epilepsy"

_ijms, 2023, doi:10.3390/ijms241914506_

Round 1

Reviewer 1 Report

The authors present an interesting study examining the influence of astrocyte activity on neuronal behaviour in the context of epilepsy. The authors utilise the PTZ model of epilepsy in establishing baseline responses to epileptic activity, before utilising an astrocyte toxin in fluorocitrate to examine changes in said baselines. Interestingly, the authors indicate that abnormal astrocyte-neuron interaction is a critical factor in epileptogenesis, and may present an interesting therapeutic target.

In reviewing the manuscript, I made a number of observations. The following should be addressed when preparing a suitable revision.

1.       The n-number for every experiments should be clearly highlighted in every legend

2.       How was the concentration of fluorocitrate decided upon for the experiments? What validation experiments were performed to indicate this appropriately interfered with astrocyte activity?

3.       For the quantitative microscopy study, how were the fields selected for measurement? Were the fields chosen at random?

4.       Can the entire Western blot be presented for review.

Author Response

Review #1 Comments and Suggestions for Authors

The authors present an interesting study examining the influence of astrocyte activity on neuronal behaviour in the context of epilepsy. The authors utilize the PTZ model of epilepsy in establishing baseline responses to epileptic activity, before utilizing an astrocyte toxin in fluorocitrate to examine changes in said baselines. Interestingly, the authors indicate that abnormal astrocyte-neuron interaction is a critical factor in epileptogenesis, and may present an interesting therapeutic target.

We very much thank the reviewer for his/her positive comments on our manuscript. In response to his/her comments, we have updated the manuscript by modifying the text where appropriate. All changes are highlighted in track changes.

  1. The n-number for every experiment should be clearly highlighted in every legend.

We now include the n-number for every experiment in the figure legend and refer to supplementary table 1.

  1. How was the concentration of fluorocitrate decided upon for the experiments? What validation experiments were performed to indicate this appropriately interfered with astrocyte activity?

Due to the wide-ranging of concentrations reported in the literature for the use of Fluorocitrate (FC) ranging from 20 to 500 micromolar, in both brain slices and intracranial injection (see below), we decide to employ a concentration of 200 micromolar based in our prior demonstration that metabolic arrest of astrocytes with thus concentration of Fluorocitrate depresses CA3–CA1 synaptic transmission, decreased Ca2+-dependent glutamate release from astrocytes, and consequently reducing the basal release probability (Bonansco et al., 2011). Importantly, this concentration has been used in subsequent studies related to the crosstalk between astrocyte and neurons (see below Vance et al., 2015; Silva-Cruz et al., 2017; Rogers et al.; 2018 and Roberts et al., 2020), indicating an appropriate concentration to interfere with astrocyte activity.

This information now is clearly stated in the material and methods section and read as follow: “In a subset of experiments, the astrocyte inhibitor Fluorocitrate (FC; 200 µM; Figure 6) [4, 42],  a reversible toxin used to decrease astrocytic metabolism [27] as inhibited aconitase that prevents the conversion of citrate to isocitrate, reducing glucose metabolism, energy stores, and cellular respiration were used. Moreover, this concentration reportedly decreases Ca2+-dependent glutamate release from astrocytes in the hippocampus [4].

References:

Different studies using FC at a concentration of 200 micromolar

  • Bonansco, C.; Couve, A.; Perea, G.; Ferradas, C. A.; Roncagliolo, M.; Fuenzalida, M., Glutamate released spontaneously from astrocytes sets the threshold for synaptic plasticity. Eur J Neurosci 2011, 33, (8), 1483-92
  • Vance KM, Rogers RC, Hermann GE. PAR1-activated astrocytes in the nucleus of the solitary tract stimulate adjacent neurons via NMDA receptors. J Neurosci. 2015 Jan 14;35(2):776-85. Doi: 10.1523/JNEUROSCI.3105-14.2015. PMID: 25589770; PMCID: PMC4293422.
  • Silva-Cruz A, Carlström M, Ribeiro JA, Sebastião AM. Dual Influence of Endocannabinoids on Long-Term Potentiation of Synaptic Transmission. Front Pharmacol. 2017 Dec 19;8:921. Doi: 10.3389/fphar.2017.00921. PMID: 29311928; PMCID: PMC5742107.
  • Roberts BM, Doig NM, Brimblecombe KR, Lopes EF, Siddorn RE, Threlfell S, Connor-Robson N, Bengoa-Vergniory N, Pasternack N, Wade-Martins R, Magill PJ, Cragg SJ. GABA uptake transporters support dopamine release in dorsal striatum with maladaptive downregulation in a parkinsonism model. Nat Commun. 2020 Oct 2;11(1):4958. Doi: 10.1038/s41467-020-18247-5. PMID: 33009395; PMCID: PMC7532441.
  • Rogers RC, McDougal DH, Ritter S, Qualls-Creekmore E, Hermann GE. Response of catecholaminergic neurons in the mouse hindbrain to glucoprivic stimuli is astrocyte dependent. Am J Physiol Regul Integr Comp Physiol. 2018 Jul 1;315(1):R153-R164. Doi: 10.1152/ajpregu.00368.2017. Epub 2018 Mar 28. PMID: 29590557; PMCID: PMC6087883.

Different studies using FC at concentrations ranging from 20 to 500 micromolar:

  • Wang T, Xu G, Zhang X, Ren Y, Yang T, Xiao C, Zhou C. Malfunction of astrocyte and cholinergic input is involved in postoperative impairment of hippocampal synaptic plasticity and cognitive function. Neuropharmacology. 2022 Oct 1;217:109191. Doi: 10.1016/j.neuropharm.2022.109191. Epub 2022 Jul 12. PMID: 35835213.
  • Vizuete AFK, Hansen F, Da Ré C, Leal MB, Galland F, Concli Leite M, Gonçalves CA. GABAA Modulation of S100B Secretion in Acute Hippocampal Slices and Astrocyte Cultures. Neurochem Res. 2019 Feb;44(2):301-311. Doi: 10.1007/s11064-018-2675-8. Epub 2018 Nov 1. PMID: 30387069.
  • Benjamin AM, Verjee ZH. Control of aerobic glycolysis in the brain in vitro. Neurochem Res. 1980 Sep;5(9):921-34. Doi: 10.1007/BF00966133. PMID: 7207696.
  • Padmashri R, Suresh A, Boska MD, Dunaevsky A. Motor-Skill Learning Is Dependent on Astrocytic Activity. Neural Plast. 2015;2015:938023. Doi: 10.1155/2015/938023. Epub 2015 Aug 4. PMID: 26346977; PMCID: PMC4539503.
  • Berg-Johnsen J, Paulsen RE, Fonnum F, Langmoen IA. Changes in evoked potentials and amino acid content during luorocitrate action studied in rat hippocampal cortex. Exp Brain Res. 1993;96(2):241-6. Doi: 10.1007/BF00227104. PMID: 7903642.
  • Cheng SC, Kumar S, Casella GA. Effects of fluoroacetate and luorocitrate on the metabolic compartmentation of tricarboxylic acid cycle in rat brain slices. Brain Res. 1972 Jul 13;42(1):117-28. Doi: 10.1016/0006-8993(72)90046-7. PMID: 5047179.
  • Clarke DD, Nicklas WJ, Berl S. Tricarboxylic acid-cycle metabolism in brain. Effect of fluoroacetate and luorocitrate on the labelling of glutamate, aspartate, glutamine and gamma-aminobutyrate. Biochem J. 1970 Nov;120(2):345-51. Doi: 10.1042/bj1200345. PMID: 5493856; PMCID: PMC1179604.
  • Paulsen RE, Contestabile A, Villani L, Fonnum F. The effect of luorocitrate on transmitter amino acid release from rat striatal slices. Neurochem Res. 1988 Jul;13(7):637-41. Doi: 10.1007/BF00973281. PMID: 2901051.
  • Saito T. Glucose-supported oxidative metabolism and evoked potentials are sensitive to fluoroacetate, an inhibitor of glial tricarboxylic acid cycle in the olfactory cortex slice. Brain Res. 1990 Dec 10;535(2):205-13. Doi: 10.1016/0006-8993(90)91602-d. PMID: 2073603.
  • Stone EA, Sessler FM, Liu WM. Glial localization of adenylate-cyclase-coupled beta-adrenoceptors in rat forebrain slices. Brain Res. 1990 Oct 22;530(2):295-300. Doi: 10.1016/0006-8993(90)91298-u. PMID: 2176116.
  • Szerb JC, Issekutz B. Increase in the stimulation-induced overflow of glutamate by fluoroacetate, a selective inhibitor of the glial tricarboxylic cycle. Brain Res. 1987 Apr 28;410(1):116-20. Doi: 10.1016/s0006-8993(87)80030-6. PMID:
  • Szerb JC, O’Regan PA. Increase in the stimulation-induced overflow of excitatory amino acids from hippocampal slices: interaction between low glucose concentration and fluoroacetate. Neurosci Lett. 1988 Mar 31;86(2):207-12. Doi: 10.1016/0304-3940(88)90572-1. PMID: 2897096.
  • Larrosa B, Pastor J, López-Aguado L, Herreras O. A role for glutamate and glia in the fast network oscillations preceding spreading depression. Neuroscience. 2006 Aug 25;141(2):1057-1068. doi: 10.1016/j.neuroscience.2006.04.005. Epub 2006 May 18. PMID: 16713108.

  1. For the quantitative microscopy study, how were the fields selected for measurement? Were the fields chosen at random?

The fields for the quantitative analysis of calcium images in astrocytes were randomly selected in the anteroposterior axis of the CA1 stratum radiatum, with astrocytes exhibiting minimum fluorescence values of 2000 units in a Look-Up Table (LUT) implemented by the Niss-Elements AR 3.2 Software (Nikon, Japan), enabling stable fluorescence measurements for at least 5 minutes.

To better clarification this information now is clearly stated in the material and methods section and read as follow: “Astrocytes were imaged using a CCD camera (Andor DR328G; Andor Technologies PLC, Ireland) attached to the microscope (Nikon, Japan) and controlled by the Niss-Elements AR 3.2 software (Nikon, Japan) that was used for offline analyses. The fields of measure were randomly selected in the anteroposterior axis of the CA1 stratum radiatum, with astrocytes exhibiting minimum fluorescence values of 2000 units in a Look-Up Table (LUT) implemented by the NISS-Elements software. Cells were illuminated with a Xenon lamp at 490 nm (200–400 ms exposure; 36,700 mm2 area). Images were acquired at 1 Hz for 5 min, regulated by a shutter (Lambda SC-Smart shutter, Sutter Instrument Company)”.

  1. Can the entire Western blot be presented for review.

The entire western blot was submitted separately for review and one additional comment for this review was made.

“I have reviewed the blots, and while I can appreciate why the blots appear cut in how they are presented in the manuscript, there were means by which the authors could have avoided this in what lanes they selected to present. For example, there are comparative lanes adjacent to one another in each blot which could have been used as the representative blot. Moreover, another concern is the lack of clarity of the examination of different regions of the hippocampus, but this is not clear in the manuscript. The authors should at minimum clarify which regions of the hippocampus are represented in the blot and utilize the lanes which do not require 'cutting' in order to present them”

In response to this additional comment regarding the blots, we now clarify the region of the hippocampus that are present in the blots (Ventral Hippocampus) in the figure legend and clearly stated it in the manuscript as follow “For protein determinations ventral hippocampal tissues from 6 animals (both Control and PTZ-Kindling) were isolated and homogenized using RIPA buffer (pH = 8.0, 150 mM, NaCl, 50 mM Tris-HCl, 1%v/v Nonidet P40, 0.1% w/v SDS, 2mM EDTA, 1.5 mM PMSF) whit protease inhibitor cocktail [Cat# G6521, Promega™])”. In addition, we use lanes that do not require the cutting to present in the main Figure 2. 

Reviewer 2 Report

The authors aimed to evaluate astrocyte and synaptic function interaction by measuring astrocytic Ca2+ activity, neuronal excitability, and excitation/inhibition balance in the hippocampus in a chronic PTZ-kindled induced epilepsy mice model. Interesting topic and good results part offering the perspective of a promising paper. Overall, this paper is written in a concise and orderly manner with sufficient introduction, detailed methods and solid data. The article is easy to read, well designated and presented, and can be of interest to reader and researchers. However, I have the following suggestions related to the improvements that should be added:

-       all references and figure legends should be written in the same format;

-  please specify the total number of animals used and how it was chosen/determined;

-       how was the 3R principle respected in the study?

-       what endpoints were taken into account to ensure animal welfare?

English is good, but there are spelling, punctuation and some grammar issues.

Author Response

Review #2 Comments and Suggestions for Authors

The authors aimed to evaluate astrocyte and synaptic function interaction by measuring astrocytic Ca2+ activity, neuronal excitability, and excitation/inhibition balance in the hippocampus in a chronic PTZ-kindled induced epilepsy mice model. Interesting topic and good results part offering the perspective of a promising paper. Overall, this paper is written in a concise and orderly manner with sufficient introduction, detailed methods and solid data. The article is easy to read, well designated and presented, and can be of interest to reader and researchers. However, I have the following suggestions related to the improvements that should be added:

We very much thank the reviewer for his/her positive and constructive comments on our manuscript. In response to his/her comments, we have updated the manuscript by modifying the text where appropriate.

-       all references and figure legends should be written in the same format.

Thanks, we have now corrected all figure legends format to be similar throughout the manuscript.

-  please specify the total number of animals used and how it was chosen/determined,

Done, the total number of animals used in this study is now stated in the material and methods section and read as follow: A total of 58 mice were divided into three experimental groups: i) the control or wild-type group (N = 19), ii) the vehicle group injected with a saline solution (1 mL/kg, intraperitoneally [i.p.], N = 14), and iii) the PTZ group (35 mg/kg, i.p., N = 25)

-       how was the 3R principle respected in the study?

Reduction: We used the minimum number of animals to obtain significant results with a 90% power and a 5% significance level.

Replacement: we encountered significant variability in the effects of PTZ, rendering mathematical or computational models’ unreliable predictors. Moreover, cell cultures proved inadequate for faithfully replicating the intricate interaction between astrocytes and neurons. In electrophysiological experiments, brain slices offered superior preservation of synaptic connections when compared to cell cultures."

The Refinement/welfare strategies are mentioned in the following question:   

what endpoints were taken into account to ensure animal welfare?

Throughout the course of this study, we observed no signs or symptoms unrelated to the anticipated outcomes of epilepsy induction that would warrant discontinuing PTZ administration in our animal model (e.g., tumors, infections, severe injuries, weight loss). As part of our welfare strategies, we maintained animals in a quiet, controlled environment with a 12/12-hour light-dark cycle and temperature control. We reduced stress by handling them for at least 15 minutes over three days before experimental procedures. Additionally, the animals were housed in enriched cages with cardboard tubes and pieces of paper for environmental stimulation. The weight of the animals was monitored daily, and we also conducted a general check of their appearance and spontaneous behavior, as well as their behavior associated with handling.

Comments on the Quality of English Language

English is good, but there are spelling, punctuation and some grammar issues.

Thanks, we have now corrected all misspelling, punctuation and grammar issues follow Native English speaker suggestions.

Round 2

Reviewer 1 Report

The authors have suitably addressed my comments